# Detecting adherence to the recommended childhood vaccination schedule from user-generated content in a US parenting forum

**Lorenzo Betti**[ID]*, **Gianmarco De Francisci Morales**[ID], **Laetitia Gauvin**[ID],
**Kyriaki Kalimeri**[ID], **Yelena Mejova**[ID], **Daniela Paolotti**[ID], **Michele Starnini**[ID]

ISI Foundation, Turin, Italy

* lrn.betti@gmail.com

**Data Availability Statement:** Due to the Terms of Use of the forum, we can not make the dataset of user posts and comments available. We release the

## Abstract

Vaccine hesitancy is considered as one of the leading causes for the resurgence of vaccine preventable diseases. A non-negligible minority of parents does not fully adhere to the recommended vaccination schedule, leading their children to be partially immunized and at higher risk of contracting vaccine preventable diseases. Here, we leverage more than one million comments of 201,986 users posted from March 2008 to April 2019 on the public online forum BabyCenter US to learn more about such parents. For 32% with geographic location, we find the number of mapped users for each US state resembling the census population distribution with good agreement. We employ Natural Language Processing to identify 6884 and 10,131 users expressing their intention of following the recommended and alternative vaccination schedule, respectively RSUs and ASUs. From the analysis of their activity on the forum we find that ASUs have distinctly different interests and previous experiences with vaccination than RSUs. In particular, ASUs are more likely to follow groups focused on alternative medicine, are two times more likely to have experienced adverse events following immunization, and to mention more serious adverse reactions such as seizure or developmental regression. Content analysis of comments shows that the resources most frequently shared by both groups point to governmental domains (.gov). Finally, network analysis shows that RSUs and ASUs communicate between each other (indicating the absence of echo chambers), however with the latter group being more endogamic and favoring interactions with other ASUs. While our findings are limited to the specific platform analyzed, our approach may provide additional insights for the development of campaigns targeting parents on digital platforms.

## Author summary

The importance and effectiveness of vaccines is generally high, but concerns toward vaccination contribute to eroding confidence in vaccination. Recently, alternative vaccination schedules are becoming popular as they allow parents to selectively delay or refuse certain vaccines depending on their specific concerns. Not being expressly anti-vaccination, these

interaction network and the Python code of the pipelines, both available at https://github.com/Loreb92/Extraction_pipelines_vaccine_hesitancy.

**Funding:** LB, LG, KK, YM and DP acknowledge financial support from the Lagrange Project of the Institute for Scientific Interchange Foundation (ISI Foundation) funded by Fondazione Cassa di Risparmio di Torino (Fondazione CRT). MS acknowledges financial support from the project Casa nel Parco (POR FESR 14/20 - CANP - Cod. 320 - 16) funded by Regione Piemonte. The funders had no role in study design, data collection and analysis, decision to publish, or preparation of the manuscript.

**Competing interests:** The authors have declared that no competing interests exist.

parents are challenging to identify on social media, however understanding the determinants of their hesitancy toward vaccines could help addressing parents' concerns through targeted interventions. In this work, we create a Natural Language Processing pipeline to automatically identify parents who state their adherence to the recommended or alternative vaccination schedule on a popular parenting forum, BabyCenter US. We find that these users have distinct interests and different experiences with vaccination, although they frequently share similar sources of information (e.g., .gov websites). Differently from what is observed on most popular digital platforms like Facebook or Twitter, where users communicate mainly with like-minded users, Babycenter users communicate between each other independently of the vaccination schedule they adopt. These observations suggest that parenting fora may be a more suitable medium to develop intervention aiming to influence positively the vaccination behavior of parents.

## Introduction

During the last decade, the USA experienced an increase of cases of vaccine-preventable diseases with frequent outbreaks of pertussis [1] and the highest number of cases of measles since 1992 [2]. Childhood immunization remains one of the most cost-effective ways to prevent the spreading of diseases, and every year the Advisory Committee on Immunization Practices revises the vaccination schedule that indicates the timing of all doses of the recommended vaccines [3]. Despite the success and safety of this strategy, parents still have concerns about specific issues related to vaccines, and more than one-third of US children are under an alternative vaccination schedule [4]. Low and partial childhood immunization coverage is associated with a higher probability of contracting vaccine-preventable diseases [5, 6], even for fully vaccinated individuals [7], and of facilitating the initial spreading of the disease [8]. In order to improve childhood vaccination uptake, a better understanding of parents' lack of trust toward vaccination is required.

Confidence in vaccination is generally high and people acknowledge its importance in preventing diseases, but concerns regarding its safety are widespread, especially in high-income countries. [9] Several conceptual models have been proposed to categorize the various factors that influence vaccination behavior [10, 11], though it has been shown that there is no strong evidence to recommend any specific strategy to reduce hesitancy and increase immunization uptake [12]. It is thus important to rely on empirical data to develop successful interventions, by understanding the specific concerns of the group under study.

The traditional tools deployed in the identification of hesitant parents and the study of their concerns include in-depth interviews and surveys, which are time-consuming and expensive. Using digital platforms such as Facebook, Twitter, or parenting fora is a compelling alternative that provides access to unprecedented amount of data in real-time [13]. There, parents use these platforms to share their health behaviors with the community and seek advice [14]. These activities generate a large amount of content from which it is possible to extract attitudes and behaviors toward vaccination, thanks to the advancements in automated data processing such as Natural Language Processing [15]. Several studies used content analysis to determine topics of discussion, narratives and vaccine-related discussion topics on digital platforms like Reddit or parenting fora [16–18]. Other studies focused on identifying users with strong opinions on vaccination (e.g., anti-vaccination and pro-vaccination) [19], but this kind of classification overlooks users with more nuanced positions who may be more suitable for intervention. Recent progresses in this direction have employed machine learning methods to

map the content of tweets to constructs of validated health behavior models [20, 21] or classify users depending on their intention to receive a specific vaccine [22]. To date, we are not aware of studies attempting to automatically determine the adherence of parents to alternative childhood vaccination schedules from user-generated content.

In this work, we study users engaging in discussion threads about vaccination on BabyCenter US, a popular parenting forum. Other studies used this source of data to investigate themes related to vaccination [23] and other topics [24], but none of them performed an analysis at the level of users. Our first contribution consists in detecting users who follow the recommended or alternative vaccination schedule by developing a high-precision classifier that searches relevant comments. Then, guided by the Determinant Matrix of Vaccine Hesitancy [10], we study how factors extracted from the activity of users on the forum are associated with the adherence to different vaccination schedules. The factors we consider include influences from personal and peer environment, such as experiences of adverse reactions following immunization (AEFI), and contextual influences, such as the sources of information cited, interests, and the way users interact with other users in the forum.

## Materials and methods

### Data collection

BabyCenter.com is a popular parenting website available in nine different languages whose contents reach 100 million people monthly [25]. There, parents or expectant parents post their experiences and ask questions, and some discuss their decision whether to vaccinate their children. Users are free to express their opinions and moderators can remove contents under specific circumstances that include hate speech, personal attacks and illegal activities [26]. These posts (and their subsequent comments) are organized into groups that revolve around a common topic of interest. In this study, we focused on BabyCenter US. To collect relevant posts, we queried the site search function with the word '*vaccine*', and downloaded all posts along with their comments. We also collected all the publicly accessible profile pages of the authors of these posts and comments, by which we obtained their self-reported location and the groups they joined.

We verified that our data collection was in compliance with the Terms of Use of BabyCenter.com [26] and we contacted the BabyCenter Community Team to inform them about our research. The Community Team forbids to interact with users and share contents on the forum. We remark that intervention and interaction activities are outside the scope of this work.

### Determining the adherence of users to vaccination schedule

To determine the adherence of users to a vaccination schedule, we examined the text of their comments, which often contain personal experiences, and disregard posts which may be phrased in a speculative or questioning manner. We developed a rule-based extraction pipeline which consists of a filter and a classifier.

The filter identifies comments which contain information related to the vaccination schedule behavior of the author. We focused on specific syntactic patterns which capture the context around a list of selected keywords (see section 2.1.1 in S1 Appendix). For example, a frequent syntactic pattern for the keyword '*schedule*' is represented in the sentence "*I* (subject) *follow* (verb) *the regular* (keyword's adjective) *vaccination* (keyword's compound) *schedule* (target keyword)". To extract syntactic relationships from sentences, we employed the dependency parser provided by the library spaCy [27]. This framework allowed us to (*i*) build a structured summary of the sentence and to (*ii*) disambiguate the context in which keywords occur to

reduce false matches (see section 2.1.1 in S1 Appendix). The output of the filter is a set of schedule-related comments.

The classifier assigns a label ('*recommended*' or '*alternative*') to schedule-related comments by taking advantage of the structured summary of the sentence. Comments are labeled as '*recommended*' by default. This label is changed whenever specific keywords related to alternative vaccination schedules are matched (e.g., '*delayed*', '*selective*') or whenever negations change the overall meaning of the comment (see section 2.1.2 in S1 Appendix). Albeit the adherence to different alternative schedule approaches may be driven by various rationales and determinants [28], we consider all of them in one class, since our methodology lacks the resolution to discern between different alternative vaccination schedules.

Seven annotators familiar with the vaccination debate evaluated the pipeline's outcome by manually labeling a sample of 300 random schedule-related comments. This set of manually labeled comments were used as test set. The annotators assigned to each comment one of three labels: *recommended*, *alternative*, or *unrelated*. We employed Cohen's Kappa $\kappa$ to measure inter-annotator agreement. The performances of the filter and the classifier were evaluated separately: first we computed the precision of the filter in the retrieval of relevant comments, then we computed the sensitivity and the specificity of the classifier after discarding unrelated comments (considering the label *alternative* as the positive class).

In the following, we focus on users who posted schedule-related comments. These users were classified as following recommended vaccination schedule (RSUs) or alternative vaccination schedule (ASUs) by aggregating all their schedule-related comments and propagating labels, after resolving conflicts due to discordant labels (see section 2.1.4 in S1 Appendix). We reported statistics for the number of RSUs and ASUs, the number of their posts, comments, and replies received. We estimated their activity period (related to vaccines) as the time elapsed between their first and last comment, after discarding users posting during a single day. We tested whether schedule-related comments written by more prolific users are more likely to be coded as *recommended* or *alternative*. To do this, we divided schedule-related comments in classes $C_{(a,b]}$, so that schedule-related comments written by users who wrote a number of schedule-related comments greater than $a$ and lower or equal than $b$ belong to the class $C_{(a,b]}$. Then, we estimated the probability and the corresponding 95% confidence interval to find a comment classified as *recommended* for each class.

## Activity of users following regular vs alternative schedule

We explored how the different views of the users on vaccination schedules are reflected in their activity on the forum. We built a set of features associated with users' activity and employed odds ratios to estimate the strength of association with the binary variable given by the adherence to a vaccination schedule. We tested the significance of the associations via Fisher's Exact Test at $p < 0.05$. The set of features analyzed are: geographical distribution, thematic groups, personal experiences of vaccine adverse reactions, specific shared content (i.e. URLs shared by users), and amount of interactions among users (in the form of comments to each other's posts). Note that geolocation and the groups followed by users can be determined from user profile pages. This limits the corresponding analysis to the set of users whose profile page is publicly accessible. The other features were extracted from users' comments.

To determine the geographic distribution of users (RSUs + ASUs) across states, we combined the self-reported geolocation of users from their profile pages with the local groups they joined to assign one of the 51 US states to each user (see section 3 in S1 Appendix). We then computed the Pearson correlation coefficient between the number of users assigned to each state and the American Census Bureau 2010 estimates [29], both log transformed.

We assumed that groups joined by users that revolve around specific topics are a proxy for their interests. To study whether the interests of RSUs and ASUs are distinct, we estimated the association between the type of vaccination schedule and each interest group.

To study how experiences of adverse events following immunization (AEFI) are associated with the decision to follow an alternative vaccination schedule, we developed an extraction pipeline able to retrieve comments mentioning experiences of AEFI. We followed the same framework discussed in the previous section (see section 2.2 in S1 Appendix). We labeled comments as *negative experience* if the author attributes the occurrence of an adverse reaction to the vaccine (e.g., *"My nephew always runs a high fever after shots"*), *positive experience* otherwise. We were interested in both first- and second-hand experiences. We evaluated this pipeline in the same way as discussed in the previous section (considering the label *negative experience* as the positive class), with the exception that in this case we manually labeled 600 comments. Finally, we labeled all authors of at least one comment classified as *negative experience* as *reporting negative experiences*, all other users are classified as *reporting positive experiences*.

Next we estimated the association between negative experiences and the adherence to the vaccination schedule. Then, in order to study if ASUs cite negative experiences as a cause to opt for an alternative vaccination schedule, we analyzed the content of the comments that were labeled by both extraction pipelines (i.e., related to both scheduling and experiences of AEFI) and authored by ASUs. We also took advantage of the structured summary of the sentences to keep track of who experienced the reaction (author of the comment, child of the author, or an acquaintance) and the kind of reaction mentioned (see section 2.2.2 in S1 Appendix). This allowed us to differentiate between first- and second-hand experiences. We evaluated the association between the vaccination schedule and both the subject of the experience and the kind of reaction mentioned.

We used URLs in the comments to study the sources of information cited by RSUs and ASUs. We reported the domains most frequently cited by RSUs and ASUs, as well as their reliability.

Finally, we checked to what extent users having different vaccination schedule behaviors interact with each other. We represented the interactions among users as a network, where each node represents a user and there is a link from user $i$ to $j$ if $i$ comments on a post by $j$. The link is directed and weighted, with weight corresponding to the number of comments of user $i$ under the posts of user $j$ (see section 4 in S1 Appendix). We quantified the homophily in the interactions by assigning each user a leaning with respect to scheduling (+ 1 for RSUs, −1 for ASUs), and computing for each user the average leaning of their neighbors, weighted accordingly to the weight of the links. We show the joint distributions of the average leaning of in- and out-neighbors separately for RSUs and ASUs.

## Results

We downloaded 54,227 posts related to vaccines and 1,129,487 comments from a total of 201,986 unique users. Only 4% of posts have no comments. The dataset spans from March 11, 2008 to April 26, 2019 (11 years and 46 days). S1 Fig shows the monthly volume of posts and comments in which two peaks of activity are visible: the first in late 2009 and the second in early 2015, probably related to the start of the H1N1 vaccination campaign and the spreading of measles linked to the Disney theme park in California respectively. In addition to the search term '*vaccine*', the retrieved posts contain also words whose stem is 'vaccin' (e.g., vaccines, vaccination). We also downloaded the user profile pages of 54% of users whose profiles were still active and public. These users joined a total of 8757 groups. We also downloaded the user

**Table 1. Statistics of the activity of recommended schedule users (RSUs) and alternative schedule users (ASUs).**

|  | $N_u$ | $N_p$ ($N_p/N_u$) | $N_c$ ($N_c/N_u$) | $N_c^s$ ($N_c^s/N_u$) | $\tau$ (IQR) | $I_p$ (IQR) |
|---|---|---|---|---|---|---|
| RSUs | 6884 | 2940 (0.427) | 123,969 (18.0) | 8613 (1.25) | 450 (838) | 11 (18) |
| ASUs | 10,131 | 5460 (0.539) | 157,008 (15.5) | 15,980 (1.58) | 400 (807) | 11 (19) |

The values correspond to the number of users $N_u$, number of posts $N_p$, number of comments $N_c$, number of schedule-related comments $N_c^s$, median time of activity $\tau$ (considering users with $\tau > 1$ day), and median number of comments under posts $I_p$.

IQR = interquartile range.

profile pages of users whose profiles were still active and public. Among them, 66,708 self-reported their geolocation and 105,795 joined at least one of 8757 groups. The comparison of the manual annotation of schedule-related comments identified by the pipeline resulted in a perfect inter-annotator agreement ($\kappa = 1.00$). S5 Fig shows the confusion matrix. The filter's performance on selecting relevant comments was considerably high, having a precision of 90.33%. The classifier also performed remarkably well, with a sensitivity of 86.43% and specificity of 95.83%.

We determined the vaccination schedule behavior of 17,015 users, 59.54% of them following an alternative vaccination schedule (see S4 Fig for the distribution of RSUs and ASUs across time). Table 1 shows the statistics related to the activity of RSUs and ASUs, and S2 Fig shows the distribution of the number of posts and comments per user, for all users, RSUs, and ASUs. On average, ASUs tend to write more posts than RSUs, whereas RSUs write more comments. However, when we consider only schedule-related comments, ASUs posted more schedule-related comments per user compared to RSUs. The median time of activity $\tau$ is greater than one year for both RSUs and ASUs, with 12% and 15% of users (respectively) being active only one day. Fig 1 shows that the estimated probabilities to find a schedule-related

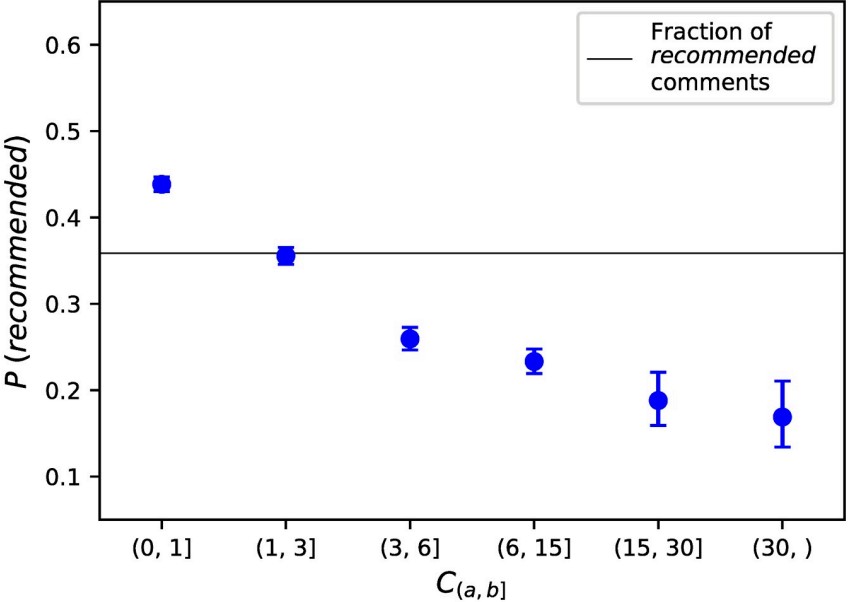

**Fig 1. Estimated probability of a schedule-related comment to be labeled as following the regular vaccination schedule for different classes.** The probability to find a schedule-related comment coded as '*recommended*' decreases for more prolific users. The horizontal line corresponds to the fraction of schedule-related comments labeled as '*recommended*'. Vertical bars correspond to 95% confidence intervals.

**Table 2. Statistics of the features extracted from the activity of recommended schedule users (RSUs) and alternative schedule users (ASUs).**

| | RSUs | | ASUs | |
|---|---|---|---|---|
| Geolocation | 2633 | 38.25% | 3814 | 37.65% |
| Groups | 2734 | 39.72% | 4332 | 42.76% |
| Experiences of AEFI | 1708 | 24.81% | 2442 | 24.10% |
| Positive experiences | 431 | 25.23%* | 356 | 14.58%* |
| Negative experiences | 1277 | 74.77%* | 2086 | 85.42%* |
| URLs | 1711 | 24.85% | 2421 | 23.90% |
| Interactions | 1004 | 14.58% | 1731 | 17.09% |

For each feature, the table shows the number of RSUs and ASUs for which the corresponding information is known. Percentages refer to fractions respect to the whole number of RSUs and ASUs.

AEFI: adverse events following immunization.

* Percentages respect to numbers of users reporting experiences of AEFI.

comment coded as '*recommended*' decreases for more prolific users: the more prolific an user, the more likely their schedule-related comments to be labeled as *alternative*. Table 2 shows the number of RSUs and ASUs for which different features are known. The proportion of ASUs ranges from 58.59% to 63.29% for each of the features.

S3 Fig shows the geographical distribution of ASUs across US states (data in S1 Spreadsheet). We found the geographical distribution of users mentioning vaccination scheduling (RSUs + ASUs) to be highly correlated with the distribution of the US population across states ($r = 0.94$, $p < 0.0001$). Nevertheless, we have no additional information to asses if this group of users is a representative sample of the US population.

We found 25 groups which are more likely to be joined by RSUs and 21 by ASUs. These groups are shown in Fig 2 with their corresponding OR and 95% confidence intervals. The group most likely to be joined by ASUs is "None/Select/Delayed Vaccinations"—a group providing support and information to parents who follow an alternative vaccination schedule. As the classification aims to detect precisely this behavior, the fact that this group is at the top of the list for ASUs proves the efficacy of the approach. ASUs are more likely to join groups related to natural and healthy lifestyles (e.g., "Home Birth—Support for Homebirth", "Organic Mamas!!", "Crunchy Mamas"), alternative methods of raising children such as the attachment parenting philosophy (e.g., "co-sleeping", "Attachment Parenting"), and homeschooling (e.g., "Homeschooling"). On the other hand, RSUs are more interested in groups such as "Formula-feeding Families", "Pumping Moms", and "Connect with Pampers (R)", which can be seen as opposed to purely "natural" practices. "Working Moms" is a group of mothers who share advice about how to reconcile professional life and being a mother.

We identified 25,634 comments related to past experiences of AEFI: 20,270 and 5364 mentioning negative and positive experiences of AEFI, respectively. The inter-annotator agreement was almost perfect ($\kappa = 0.84$), and the performance of the pipeline was satisfying, with precision of 92.83%, sensitivity of 94.15%, and specificity of 71.54%. S5 Fig shows the confusion matrix. Through these comments, we labeled a total of 1708 RSUs and 2442 ASUs. Compared with RSUs, ASUs are about twice as likely to report negative experiences of AEFI (OR = 1.98, 95% CI: 1.69–2.31). There are 1152 comments authored by ASUs and classified by schedule and AEFI pipelines (85% of them reporting negative experiences of AEFI). Content analysis showed two different directions of causality as shown in Table 3. Both suggest that the adherence to an alternative vaccination schedule may be considered as a strategy to compensate for the perceived risks of vaccines, and positive experiences reinforce this belief. ASUs and RSUs

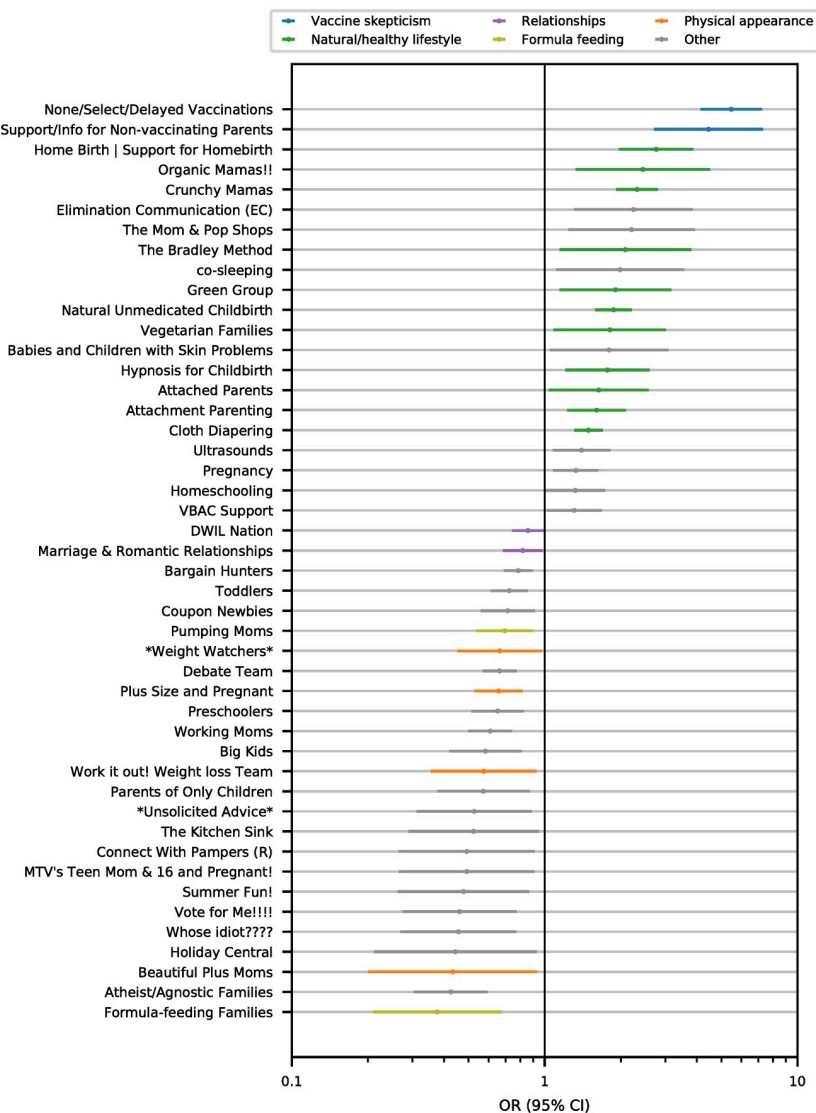

**Fig 2. Interests of recommended schedule users and alternative schedule users.** Odds ratio (OR), on a logarithmic scale, of being a member of the group given the vaccination schedule followed by the users (recommended or alternative) for the groups whose OR differs from 1 with confidence level of 95% (horizontal bars correspond to 95% confidence intervals). The groups are sorted by decreasing values of OR from the most closely associated with alternative schedule users (top) to the most closely associated with recommended schedule users (bottom). Colors refer to different categories of groups.

are equally likely to mention first-hand experiences (author's experience OR = 1.08, 95% CI: 0.92–1.28; child's experience OR = 1.01, 95% CI: 0.87–1.17) while ASUs are about three times more likely to mention second-hand experiences (OR = 3.18, 95% CI: 2.02–5.00). Fig 3 shows that ASUs are more likely to report more serious adverse reactions such as seizure and developmental regression, while RSUs are more likely to report common reactions as fever, pain, and fussiness.

We found 5213 comments written by RSUs and 7314 by ASUs to contain URLs. Fig 4 shows the most frequent domains shared by RSUs and ASUs, with horizontal bars indicating separately the fraction of comments by RSUs and ASUs containing the corresponding domain. Besides the domain corresponding to the BabyCenter forum, whose links point to other posts

**Table 3. Causality between vaccination schedule adopted and experiences of adverse events following immunization.**

| | Label of comments | Frequency |
|---|---|---|
| *The author states that adverse reactions lead him/her to switch to an alternative schedule* | Negative experience | 56/100 |
| *The author states that following an alternative schedule contributed to the lack of adverse reaction after the vaccine* | Positive experience | 26/100 |

The table shows two sentences encoding the corresponding directions of causality identified and their occurrence into the sample of comments analyzed. The two direction of causality was found in comments having different labels respect to experiences of AEFI: the first in comments labeled as *negative experience* and the second in comments labeled as *positive experience*.

or groups of the forum, the top ranked domains for both RSUs and ASUs point to governmental agencies, namely the Center for Disease Control and Prevention (cdc.gov), the National Center for Biotechnology Information (ncbi.nlm.nih.gov, whose URLs mainly point to research articles), and the Food and Drug Administration (fda.gov). After the most frequently shared domains, there are also domains mainly preferred by one of the two groups of users. The National Vaccine Information Center (nvic.org) and the website AskDrSears.com support skeptic views on vaccinations and advocate alternative vaccination schedules and they are indeed favored by ASUs. Differently, the journal of American Academy of Pediatrics (pediatrics.aappublications.org), the website of the Children Hospital of Philadelphia, and the website

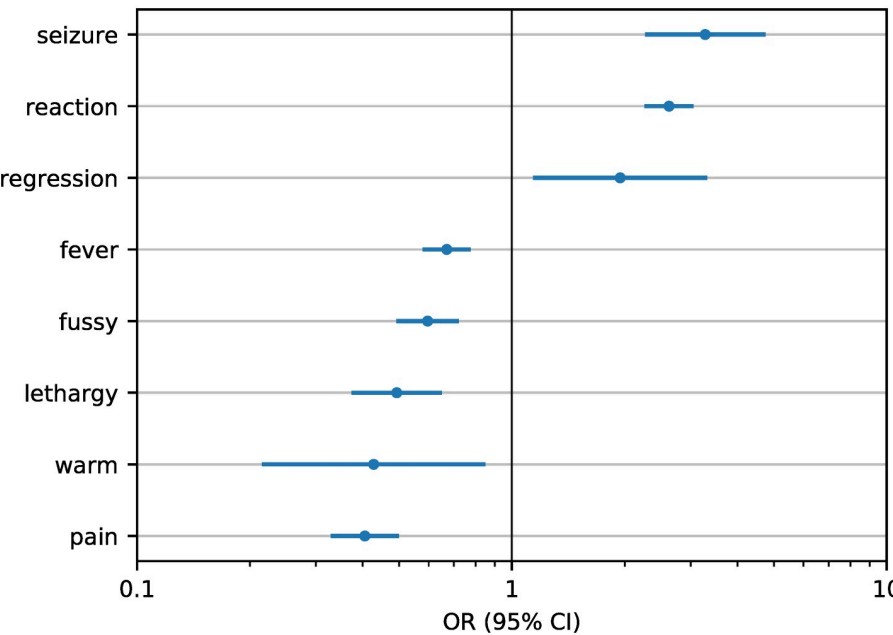

**Fig 3. Kind of adverse reaction mentioned by recommended schedule users and alternative schedule users.** Odds ratio (OR), on a logarithmic scale, of reporting a specific adverse reaction following immunization given the vaccination schedule followed by the users (regular or modified). Reactions whose OR differs from 1 with confidence level of 95% (horizontal bars correspond to confidence intervals) are reported and sorted by decreasing values of OR from the most closely associated with alternative schedule users (top) to the most closely associated with recommended schedule users (bottom).

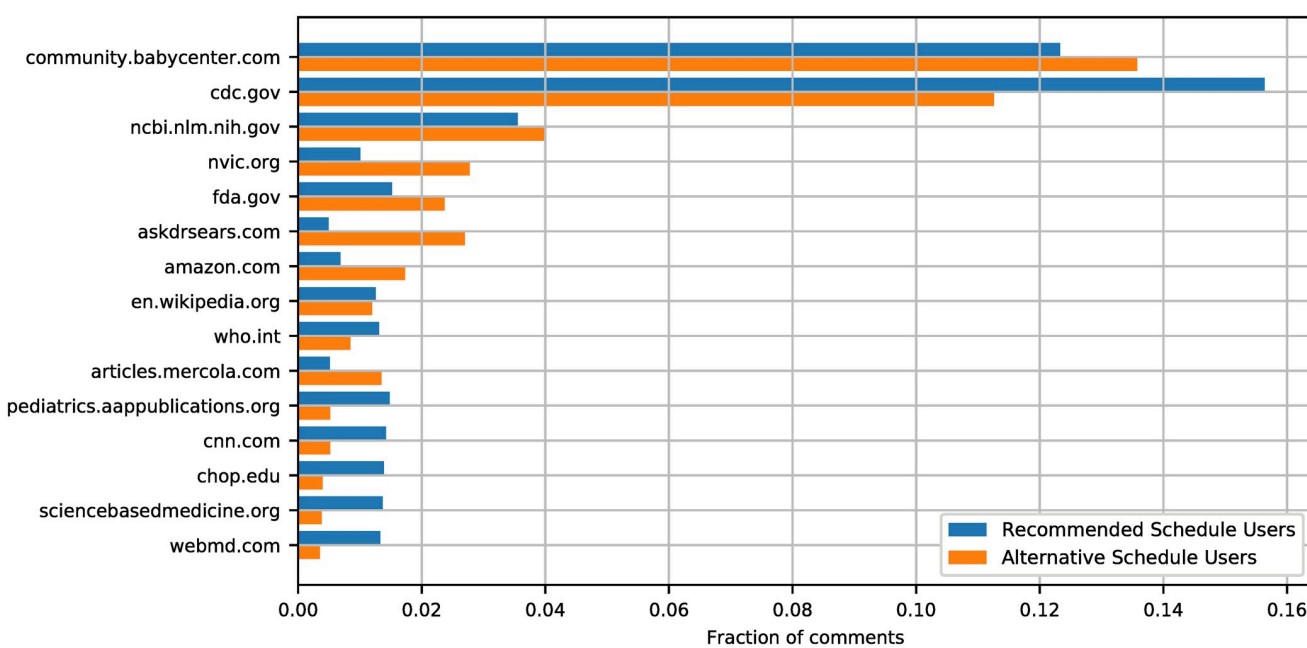

**Fig 4. Top 15 domains shared by recommended schedule users (RSUs) and alternative schedule users (ASUs).** Domains are sorted by total occurrence in comments. The length of the bars corresponds to the percentage of comments of RSUs (blue) and ASUs (orange) containing the corresponding domain.

Science-Based Medicine (sciencebasedmedicine.org) are preferred by RSUs as they firmly support the recommended vaccination schedule.

The network includes 234,815 interactions between RSUs and ASUs. Fig 5 shows contour maps of the joint probability distributions of the average leaning of in- and out-neighbors, separately for RSUs (left) and ASUs (right). Both distributions are skewed toward negative values of the average leaning of in-neighbors, indicating a higher interest toward posts submitted by ASUs. However, the leaning of users commenting on their posts is different for the two groups: whereas it is skewed to −1 for ASUs (right graph), it is distributed more evenly for RSUs (left graph). ASUs have almost twice as many comments in our dataset, which may account for some of the attention skew, but despite this bias we find RSUs to be actively engaged with posts from both sides. These observations paint a picture far from what we would expect in presence of an echo chamber, which would imply a joint probability distribution peaked around (+ 1, + 1) for RSUs and around (−1, −1) for ASUs. Despite the lack of echo chambers, homophily is not completely absent: while RSUs display interest toward the contents produced by ASUs, ASUs behave as an endogamic group both paying attention to and having comments from their own group.

## Discussion

In this work, we automatically detected the vaccination schedule adherence of parents on an online parenting forum. Thus, we fill the methodological gap between detailed but smaller-scale traditional survey- or interview-based studies and the large-scale internet studies identifying potentially over-simplified pro- or anti-vaccination attitudes [30, 31].

Our automated pipeline identified 17,015 users discussing vaccination scheduling with sensitivity of 86.43% and specificity of 95.83% (while the intermediate step of finding relevant comments performed at 90.33% precision). Out of these, 60% expressed an intention to follow an alternative vaccination schedule (ASUs). This group is more vocal than RSUs when writing

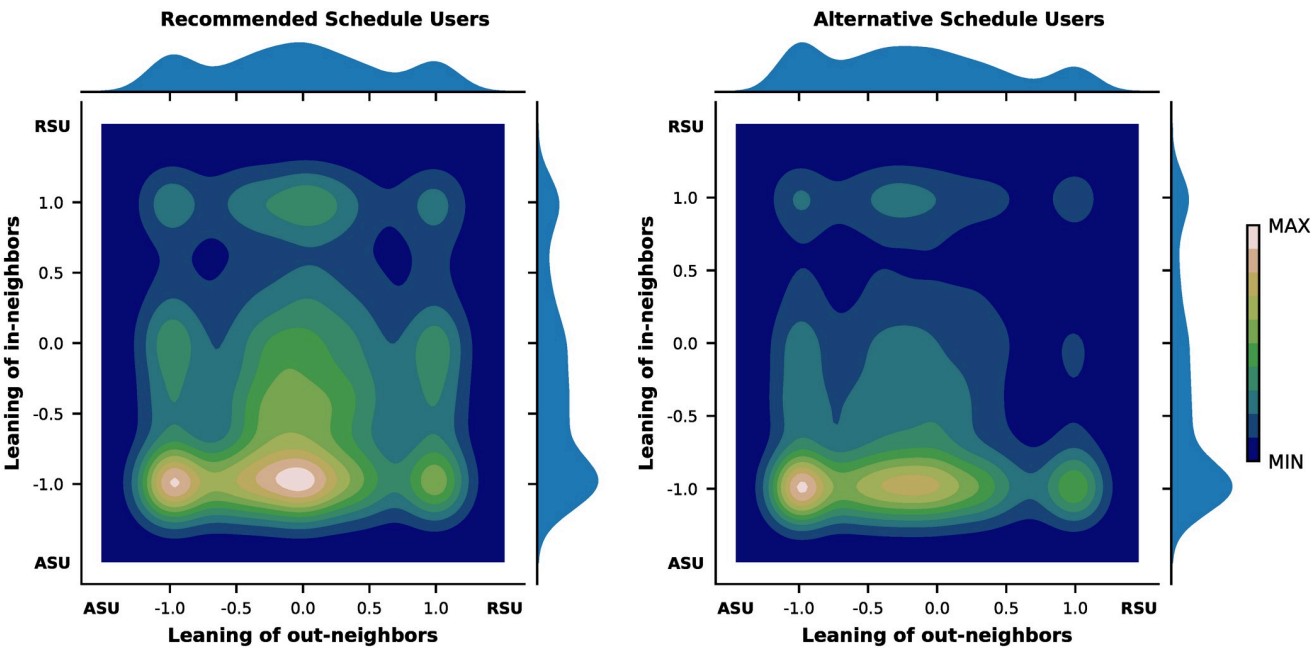

**Fig 5. Joint distribution of the average leaning of in- and out-neighbors of recommended schedule users (RSUS) and alternative schedule users (ASUs).** Figures refer to RSUS (left) and ASUs (right). The color bar presents the scale of colors, from lower (dark blue) to higher values of probability (white).

about vaccination schedule (see Fig 1) and this may account for a possible underestimation of the number of RSUs. On the other hand, this may be a potential benefit for intervention campaigns as it is easier to identify users following an alternative vaccination schedule.

Guided by the framework of the Determinant Matrix of Vaccine Hesitancy [10], we examined how the adherence to different vaccination schedules is associated with factors identified in previous literature. We found that ASUs are more likely to be interested in alternative medicine and natural lifestyles, suggesting a broader distrust of healthcare providers and mainstream medicine [32–34]. Interestingly, despite some questionable beliefs that vaccines may cause diseases, we found that RSUs and ASUs are equally likely to join groups focused on support for parents with children having particular medical conditions (e.g., autism spectrum disorder). Furthermore, we found that ASUs are more likely than RSUs to report negative experiences of AEFI (OR = 1.98, 95% CI: 1.69–2.31) which suggests the adherence to alternative schedules is a response to safety concerns [35]. In fact, our results support previous observations that second-hand experiences may influence vaccination behavior (OR = 3.18, 95% CI: 2.02–5.00) [35], as well as the severity of the reaction experienced (e.g., seizure OR = 3.28, 95% CI: 2.26–4.75). Thus, we are able to corroborate survey-based results via a large-scale automated analysis of user-generated data.

Moreover, we obtained two surprising observations encouraging the use of platforms like BabyCenter for opinion change campaigns. Unlike on other digital platforms like Facebook or Twitter where communication between anti- and pro-vaccination camps is minimal (resulting in a strong echo chamber phenomenon [30, 31]), we found the communication between RSUs and ASUs to be more unobstructed on this parenting forum. In addition, a potential middle ground may come in the form of official governmental resources, as both frequently share materials from ".gov" domains. Previous studies found that hesitant parents acknowledge the importance of immunizations [36], but consider health care professionals as biased toward

mainstream medicine. They thus turn to other sources of information, both official and personal [37], and subsequently face an information synthesis problem which may encourage a hedging approach of alternative vaccination scheduling [36]. A clear communication by the governmental entities about the intricacies of parent concerns may thus reach skeptical parents and provide a common ground of scientifically established facts.

Despite encouraging findings, our method should be used in combination with traditional methodologies, as automated internet analysis has several notable limitations. It is not likely this specific platform provides a representative subset of U.S. parents. Notwithstanding an high correlation of the distribution of users and US population across states, we have no information about our sample's socio-demographic representativeness. In addition, we have no information about users who read the content published on the forum without participating, another group of individuals who may benefit from intervention campaigns. This platform comes with its specific normative biases likely different from other internet platforms, whereas methodological choices of keywords and filtering rules further constrain the studied sample. Focusing on the precision of our pipeline by hand-crafting keywords and syntactic rules, we may miss expressions of vaccine hesitancy which are more difficult to detect automatically (properly assessing the recall of our system would mean manual annotation of a substantial portion of the dataset). However, our methodology can be customized both for other sources of data, by modifying the lists of keywords, and domains, as we illustrated by adapting the vaccination schedule pipeline to identify and classify experiences of AEFI. Our models may be improved using deep-learning techniques [38] that may produce models both more generalizable to new domains and flexible in terms of different vaccination hesitancy stances (although, again, this may need an extensive annotation effort). These limitations, however, complement those of traditional survey-based methods. For example, it is possible to capture ongoing discussions which happen among fellow parents, potentially reducing recall, motivated forgetting and conformity biases. Moreover, topics of discussion are not limited by a surveyor's predetermined set of hypotheses.

Thus, we argue that our general framework can pave the way for a proactive approach to engage with large populations open to debate and new information. In that sense, we propose that the standard vaccination hesitancy communications playbooks, such as the ones developed by the WHO [39] and CDC [40] are extended to include parents (instead of the general public) seeking specific information that is not over-simplified and which incorporate the latest scientific findings (instead of simple messages based on hashtags that are the current norm). Already, there is evidence showing that interventions based on websites combining vaccine information and interactive social media components can influence positively the behavior of parents [41]. Similarly, other experiments show the ability to correct health-related misconceptions also of users who do not directly participate in the discussion on social media (a phenomenon called "observational correction") [42, 43]. We encourage the incorporation of high-precision automated tools for targeted intervention campaigns to counter vaccine hesitancy, as a coordinated effort with health care professionals and policy makers to develop effective and non-intrusive strategies with which users are comfortable.

## Supporting information

**S1 Appendix. Supplementary appendix.** This document provides supplementary information regarding: data collection and structure of the dataset (section 1), extraction pipelines (section 2), geolocation of users (section 3) and construction of the interaction network (section 4). (PDF)

**S1 Fig. Monthly volume of posts and comments related to vaccination on BabyCenter US.** Number of posts (red) and number of comments (blue).
(PDF)

**S2 Fig. Distribution of the number of posts per users (left) and number of comments per user (right).** The distributions for all users, regular schedule users (RSUs) and alternative schedule users (ASUs) are shown.
(PDF)

**S3 Fig. Distribution of the fraction of alternative schedule users (ASUs) across US states.** The authors created the map based on the built-in geometry of the open source Python library Plotly (https://plotly.com/python/).
(PDF)

**S4 Fig. Number of users who joined the forum yearly.** The solid line indicates the number of users who joined the forum yearly, while vertical bars show the distribution of regular schedule users (RSUs) and alternative schedule users (ASUs) among new users.
(PDF)

**S5 Fig. Confusion matrices of the vaccination schedule pipeline (left) and experiences of AEFI pipeline (right).** Raw numbers are shown within each cell.
(PDF)

**S1 Spreadsheet. Number of recommended schedule users (RSU) and alternative schedule users (ASU) mapped to US states.**
(XLSX)

## Author Contributions

**Conceptualization:** Lorenzo Betti, Gianmarco De Francisci Morales, Laetitia Gauvin, Kyriaki Kalimeri, Yelena Mejova, Daniela Paolotti, Michele Starnini.

**Data curation:** Lorenzo Betti, Gianmarco De Francisci Morales, Laetitia Gauvin, Kyriaki Kalimeri, Yelena Mejova, Daniela Paolotti, Michele Starnini.

**Formal analysis:** Lorenzo Betti.

**Funding acquisition:** Lorenzo Betti, Laetitia Gauvin, Kyriaki Kalimeri, Yelena Mejova, Daniela Paolotti, Michele Starnini.

**Investigation:** Lorenzo Betti, Gianmarco De Francisci Morales, Laetitia Gauvin, Kyriaki Kalimeri, Yelena Mejova, Daniela Paolotti, Michele Starnini.

**Methodology:** Lorenzo Betti, Gianmarco De Francisci Morales, Laetitia Gauvin, Kyriaki Kalimeri, Yelena Mejova, Daniela Paolotti, Michele Starnini.

**Project administration:** Gianmarco De Francisci Morales, Laetitia Gauvin, Kyriaki Kalimeri, Yelena Mejova, Daniela Paolotti, Michele Starnini.

**Resources:** Gianmarco De Francisci Morales, Laetitia Gauvin, Kyriaki Kalimeri, Yelena Mejova, Daniela Paolotti, Michele Starnini.

**Software:** Lorenzo Betti.

**Supervision:** Gianmarco De Francisci Morales, Laetitia Gauvin, Kyriaki Kalimeri, Yelena Mejova, Daniela Paolotti, Michele Starnini.

**Validation:** Lorenzo Betti, Gianmarco De Francisci Morales, Laetitia Gauvin, Kyriaki Kalimeri, Yelena Mejova, Daniela Paolotti, Michele Starnini.

**Visualization:** Lorenzo Betti.

**Writing – original draft:** Lorenzo Betti, Gianmarco De Francisci Morales, Laetitia Gauvin, Kyriaki Kalimeri, Yelena Mejova, Daniela Paolotti, Michele Starnini.

**Writing – review & editing:** Lorenzo Betti, Gianmarco De Francisci Morales, Laetitia Gauvin, Kyriaki Kalimeri, Yelena Mejova, Daniela Paolotti, Michele Starnini.

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
