## [Decision Letter · Decision Letter 0]

6 Jan 2021

Dear Dr. Betti,

Thank you very much for submitting your manuscript "Detecting adherence to the recommended childhood vaccination schedule from user-generated content in a US parenting forum" for consideration at PLOS Computational Biology.

As with all papers reviewed by the journal, your manuscript was reviewed by members of the editorial board and by several independent reviewers. In light of the reviews (below this email), we would like to invite the resubmission of a significantly-revised version that takes into account the reviewers' comments.

We cannot make any decision about publication until we have seen the revised manuscript and your response to the reviewers' comments. Your revised manuscript is also likely to be sent to reviewers for further evaluation.

Sincerely,

Roger Dimitri Kouyos

Associate Editor

PLOS Computational Biology

Virginia Pitzer

Deputy Editor

PLOS Computational Biology

Reviewer's Responses to Questions

**Comments to the Authors:**

Reviewer #1: Thank you for the opportunity to review this manuscript, which seeks to characterise and compare two types of users. The strategy for selecting users from two groups is a reasonable one (but isn't as reflected in the results as it could be), the methods are generally transparent and replicable, but there are some minor issues with the framing and the attempt to draw conclusions about the implications of the work.

Overall: As with all other studies of individual online forums, the participants are self-selected and represent a special subgroup of people. Some of those people are over-represented as users in online forums. As a consequence any data analysis is fundamentally limited by an obscured view of a very biased sample of the population and this limits the conclusions that can be drawn. Suggesting that data from a forum can be used to support the development of digital interventions is problematic because there is an implicit assumption that (a) the results will generalise over time and beyond the forum, and (b) that people who openly discuss or advocate for ASUs are the right "target" for digital interventions (people reading without participating are more likely to benefit from interventions - more below).

Abstract: Nice to see an abstract that includes details about the dataset, what was done to it, and clear description of findings. The only issue was the sentence starting with "Though", which does not read well. As above, I also disagree with the implications sentence and I think it can be dangerous to infer from a special subset of people who post on forums about what the population thinks and feels.

Author Summary: I do agree with the last part of this section and I think this is a more nuanced statement around the capacity to enhance communication between people who promote (or simply make) different decisions. Is there any risk that this will backfire and increasing communication will instead increase hesitancy and refusal where certain groups become more persuasive for the "audience" (not the people posting but the people reading the posts)?

Introduction: It wasn't clear why the focus was only on the US when this is a global problem and I assume the forum is not restricted to US posters and US audiences?

Introduction: Very nice to see that the work is guided by a conceptual framework rather than something simple or something reinvented.

Data Collection: Typo "Babycenyer"

Data Collection: I read the statement that the data could not be released to the public but the method of data collection seems to suggest the website was crawled. This would be the appropriate place to note that this was aligned with the terms of service for the website, or that permission was granted, and evidence that human ethics was granted or waived by the university. I would not want to be PLOS Computational Biology if a user reads the paper and decides they see themselves in the data if it hasn't at least been checked by an ethics committee.

Determining adherence: Nice to see that multiple independent annotators were used and they were blinded to the output of the classifier label to check for correctness.

Activity of users: I think it is potentially dangerous to look at geographic representativeness. While it isn't a bad thing on its own, it starts to make it look like the work is headed towards a "representative sample" of the United States. There are so many reasons why people who decide to post on a forum are never going to be representative. Looking at location might give readers a false sense of representativeness that absolutely does not exist. It would be much better to look at the representativeness of the "audience" and not the users posting.

(As evidence, 201K users posted comments over 11 years but the website supposedly reaches 100 million people; which means you get a very biased 0.2% of the audience. It makes it much worse that ASUs are heavily over-represented on the forum when this is not at all reflected in the population. Clearly people promoting vaccine choice/hesitancy/etc. are much more vocal.)

Results: the first does not read well "96% of which with at least one comment,"

Results: The way the study is set up is that two groups of users were to be compared in terms of their behaviour on the site, but then the results are mostly about characterising the whole group.

Results: The groups analysis and Figure 2 are very nice and likely "actionable" in the sense of understanding where audience/users are at higher or lower risk.

Discussion: :"Out of these, 60% expressed an intention to follow an alternative vaccination schedule (ASUs) – a much higher fraction than the one estimated by surveys [26]. This may be due to ASUs being more vocal about their scheduling choices (see Figure 1)." This is simply the wrong comparison, or perhaps just the wrong wording. Obviously the proportion of users commenting about schedules on a forum bears no resemblance to the underlying distribution or clustering of hesitancy in the population and the comparison doesn't make sense - instead it might be worth including both numbers in the same paragraph to say clearly that people promoting ASUs are vocal on this particular forum and this is obvious because it is completely unaligned with prevalence.

Discussion: The finding that there is frequent communication between the groups is interesting and worth focusing on.

Discussion: The note about representativeness buried in the discussion is not enough in my opinion. Readers will ignore this limitation when citing or describing this work.

Reviewer #2: In this work the authors analyse posts extracted from a public online forum (babycenter.com), on which current or prospective parents exchange information about childcare. The authors used a Natural Language Processing approach to detect posts related to vaccinations and determined whether these posts were by parents following a recommended (RSU) or alternative (ASU) vaccination schedule for their children. The authors also determine whether the posts are about specific adverse effects of vaccinations. They then compare RSU and ASU groups for differences in the type of content they share and find that ASUs are more likely to focus on alternative medicine and mention more severe adverse effects. They also explore the presence of echo chambers on the site and find that the two groups communicate with each other, however ASUs express stronger homophily. The work ends with specific recommendations for targeted public health interventions on online media, such as forums.

I find the work interesting as it gives insight on the concerns regarding vaccinations by parents. By analysing a relatively large dataset in an automated way, the authors are able to confirm several findings from surveys. As the work shows, online forums can therefore be an interesting public data source for answering specific questions regarding attitudes towards vaccination. My comments are mostly minor:

* The authors state that 7 annotators have manually labelled 300 random comments and proceed to implement a fairly complex rule-based classifier. The authors don't mention a split into a validation and leave-out test set. If the authors used the same set for validation and testing then I would expect the performance scores to be overestimated, as complex rule-based algorithms have a tendency to overfit. 

* The label distribution for the manually annotated data is not given. Is there label imbalance?  

* Similarly, performance scores should be provided for all estimated classes. 

* "RSUs posted more schedule-related comments per user compared to ASUs" (line 173). This seems to contradict the data in table 1 and the discussion (line 258). 

* I'm missing some information regarding content moderation. Are extremist views removed from the site? If so, under which circumstances. 

* I have a hard time understanding the long sentence line 308-312. Maybe should be split into multiple sentences. 

* The term "hashtag-able" might not be understood by all readers (line 319). 

* "Differently from what is observed on most popular digital platforms like Facebook or Twitter, where users communicate only with like-minded user" (author summary). "Only" is too strong here. 

* "We downloaded 54,227 posts related to vaccines, 96% of which with at least one comment, and 1,129,487 comments from a total of 201,986 unique users." (line 162/163) - a bit hard to read (again, should be split into two sentences). 

* The dataset spans more than 11 years. Could a temporal trend be observed? Could this data source be used for monitoring? 

* The introduction doesn't cover whether studies exist in which similar forums have been analysed. I would also count Reddit or Gab as more discussion-based forums in this context.  

* Although it is understandable that the raw data cannot be made public, I was wondering whether certain aggregated datasets could be provided. In particular the interaction network between (Fig. 5) could be of interest. 

* The choice of the singular search term "vaccine" (and not a more complex regex pattern) is a bit unfortunate - however the authors have already partially hinted at this issue in the discussion (line 297). 

* I've had difficulties following the term "class length" (line 178). If I understood correctly this is simply the number of schedule-related comments. Maybe something more descriptive than "class length" could be used in the text. Also, Figure 1 is missing the corresponding x-axis label. 

* What does the horizontal line in Fig. 1 mean? Is it simply the mean over all data?

Reviewer #3: The authors present an analysis of data collected on a popular parenting forum, to detect adherence to recommended vaccination schedules among users and identify factors associated with it. Below are my comments:

- l78: "comments are labeled as `recommended`by default". Can the method be applied with no "default" label, and check how many patients are then labeled in the ASU and RSU groups? Or in other terms, what is the proportion of users that clearly state which schedule they follow, and can we reasonnably assume that the ones not clearly stating it belong to the RSU group?

- l165: "54% of users whose profiles were still active and public". It is not completely clear to me in which part of the analysis the full data of all users was used and in which part it was only reduced to the 54% subset. Is the distribution of ASU and RSU in this subset similar to the one in all of the users?

- description of the ASU and RSU data: I think some additional points could be added to the descritive analysis to the data. In particular, if the geolocation of some users is available, could we have an idea of how ASU vs RSU are localized in the US? In addition to Fig 1, could we see the distribution of the number of comments per user, among all users and then ASU vs RSU? It would give some insights in term of behavior of users.

In addition, do you have access to other user-specific variables, such as gender or age?

For users, can you obtain the date at which they joined the forum, and if yes, is it possible to see the distribution of date at which users joined, among all users and ASU vs RSU? Did more ASU join in the more recent years?

- Fig2: instead of looking at each group separately (as it is a bit hard to make sense of all of the OR with only the name of the groups), would it be possible to create categories of groups here (e.g., "natural and healthy lifestyle" would include users in at least one of the following group: Home Birth, Organic Mamas..)?

- network of users interactions: how does the network adjust for the different distributions of comments among RSU and ASU, as it seems that ASU have higher number of comments that RSU overall?

- l254 to 258: should be in the results section?

- could you use your method to identify comments related to specific diseases, such as measles and pertussis, as mentioned in the introduction? If yes, could we identify for example the temporal dynamics of the number of comments about measles by ASU and RSU and look if the number of comments increase in the last years (with the raise of cases in the US)?

**Have all data underlying the figures and results presented in the manuscript been provided?**

Reviewer #1: **No: **The authors note that the website does not provide these data but there was some inconsistency in terms of how the data were accessed in the manuscript. If the data are protected, then I would imagine they would need human ethics as well.

Reviewer #2: **No: **The study is based on data which cannot be shared.

Reviewer #3: None

PLOS authors have the option to publish the peer review history of their article (what does this mean?). If published, this will include your full peer review and any attached files.

Reviewer #1: No

Reviewer #2: **Yes: **Martin Müller

Reviewer #3: No
---

## [Decision Letter · Decision Letter 1]

11 Mar 2021

Dear Dr. Betti,

Thank you very much for submitting your manuscript "Detecting adherence to the recommended childhood vaccination schedule from user-generated content in a US parenting forum" for consideration at PLOS Computational Biology. As with all papers reviewed by the journal, your manuscript was reviewed by members of the editorial board and by several independent reviewers. The reviewers appreciated the attention to an important topic. Based on the reviews, we are likely to accept this manuscript for publication, providing that you modify the manuscript according to the review recommendations.

Sincerely,

Roger Dimitri Kouyos

Associate Editor

PLOS Computational Biology

Virginia Pitzer

Deputy Editor-in-Chief

PLOS Computational Biology

[LINK]

Reviewer's Responses to Questions

**Comments to the Authors:**

Reviewer #1: Thank you for the opportunity to review the revised manuscript, which has improved substantially from the original submission. I have been through the responses and the changes are generally appropriate, and below I have made comments only on the changes that were incomplete or may be inadequate and may remain as issues in this revised manuscript.

On "audience" (comments and responses were not numbered), it is worth looking through the work of Bode and Vraga (and cited/citing work) in the last three years, who have worked on several experiments related to the effect on the "audience" for interventions on social media.

The comment on the release of data and the use of public data seem to be in conflict. If the data are exempt from IRB approval (which I don't believe was sought) because they are in the public domain and can be accessed, then why can't the data also be made available so that readers can independently assess the robustness of the claims made in the work? Essentially this is saying: "There were no ethical requirements for not sharing the data and the data are public, but we won't provide the snapshot of data we collected from the website with readers to allow them to verify our claims." Either there are ethical reasons why the data can't be shared or the data should be shared.

The Second Reviewer's comment on overfitting was not addressed.

Reviewer #2: The authors have addressed all my comments.

Reviewer #3: I believe the authors addressed a substantial amount of my comments to a satisfactory level and I do not have any additional comments.

There are still a couple of typos:

- l213-214: ranges from.. to

- l242: should be S5Fig and not S2Fig

**Have all data underlying the figures and results presented in the manuscript been provided?**

Reviewer #1: **No: **

Reviewer #2: **No: **Some of the data cannot be made public

Reviewer #3: Yes

PLOS authors have the option to publish the peer review history of their article (what does this mean?). If published, this will include your full peer review and any attached files.

Reviewer #1: No

Reviewer #2: No

Reviewer #3: No

Figure Files:

Data Requirements:

Reproducibility:

References:

---

## [Editor Report · Decision Letter 2]

26 Mar 2021

Dear Dr. Betti,

We are pleased to inform you that your manuscript 'Detecting adherence to the recommended childhood vaccination schedule from user-generated content in a US parenting forum' has been provisionally accepted for publication in PLOS Computational Biology.

Best regards,

Roger Dimitri Kouyos

Associate Editor

PLOS Computational Biology

Virginia Pitzer

Deputy Editor-in-Chief

PLOS Computational Biology

---

## [Editor Report · Acceptance letter]

12 Apr 2021

PCOMPBIOL-D-20-02123R2 

Detecting adherence to the recommended childhood vaccination schedule from user-generated content in a US parenting forum

Dear Dr Betti,

I am pleased to inform you that your manuscript has been formally accepted for publication in PLOS Computational Biology. Your manuscript is now with our production department and you will be notified of the publication date in due course.

With kind regards,

Andrea Szabo
